# Retinal OCTA Image Segmentation Based on Global Contrastive Learning

**DOI:** 10.3390/s22249847

**Published:** 2022-12-14

**Authors:** Ziping Ma, Dongxiu Feng, Jingyu Wang, Hu Ma

**Affiliations:** 1College of Mathematics and Information Science, North Minzu University, Yinchuan 750021, China; 2College of Computer Science and Engineering, North Minzu University, Yinchuan 750021, China

**Keywords:** medical image processing, image segmentation, optical coherence tomography angiography, retinal vascular plexus, imbalanced data, contrastive learning, convolutional neural network

## Abstract

The automatic segmentation of retinal vessels is of great significance for the analysis and diagnosis of retinal related diseases. However, the imbalanced data in retinal vascular images remain a great challenge. Current image segmentation methods based on deep learning almost always focus on local information in a single image while ignoring the global information of the entire dataset. To solve the problem of data imbalance in optical coherence tomography angiography (OCTA) datasets, this paper proposes a medical image segmentation method (contrastive OCTA segmentation net, COSNet) based on global contrastive learning. First, the feature extraction module extracts the features of OCTA image input and maps them to the segment head and the multilayer perceptron (MLP) head, respectively. Second, a contrastive learning module saves the pixel queue and pixel embedding of each category in the feature map into the memory bank, generates sample pairs through a mixed sampling strategy to construct a new contrastive loss function, and forces the network to learn local information and global information simultaneously. Finally, the segmented image is fine tuned to restore positional information of deep vessels. The experimental results show the proposed method can improve the accuracy (ACC), the area under the curve (AUC), and other evaluation indexes of image segmentation compared with the existing methods. This method could accomplish segmentation tasks in imbalanced data and extend to other segmentation tasks.

## 1. Introduction

Optical coherence tomography angiography (OCTA), a non-invasive imaging technique, has been increasingly utilized for imaging the retinal and choroidal vascular system at capillary-level resolution. Compared with previous angiographic methods, OCTA has the merits of easier handling, faster imaging, and greater imaging depth, thus facilitating quantitative assessment of the morphological structure of the retinal and choroidal vascular system. Abnormalities on OCTA images usually indicate the presence of a number of diseases, such as early glaucomatous optic neuropathy [1], diabetic retinopathy [2], and age-related macular degeneration [3]. Some other recent studies have shown that deep microvascular morphological changes revealed by OCTA images are associated with Alzheimer’s disease [4] and mild cognitive impairment [5]. This opens up new ideas to investigate the relationship between abnormalities in retinal vessels and various neurodegenerative diseases. This shows that the precise segmentation of retinal vessels, especially deep capillaries, is of great importance to clinicians.

Before the popularity of deep learning, most of the automatic vessel segmentation methods were based on filters, and such methods are generally referred to as traditional methods. Azzopardi et al. [6] designed a rod structure selection filter based on the COSFIRE filter, which can effectively extract information about branching structure compared to other filters. A detailed review of the methods for conventional vessel detection and segmentation can be found in the literature [7]. A common limitation of all these traditional vessel segmentation methods is that they usually require manual tuning of parameters for a specific dataset, which makes the algorithms less scalable.

Compared to traditional methods, deep learning performs better generalization. However, deep learning methods applied to medical images still face challenges, which Zhou et al. [8] attribute to the characteristics of medical images itself: the lack of standardized big data; the multimodality of medical data; and the long-tailed distribution of morbidity. Such problems also exist in the field of ophthalmic images, and more detail can be found in earlier work [9,10,11,12,13]. In previous retinal vascular datasets, the number of samples is generally low, such as in DRIVE [14], CHASE-DB1 [15], etc. To alleviate the shortage of such datasets, Li et al. [16] released the largest multimodal OCTA image dataset, OCTA-500, and also proposed the IPN image projection network. Taking advantage of the vertical consistency of 3D OCTA images, the IPN method compresses information in a 3D image to a 2D image and finally outputs segmentation maps. Ma et al. [17] constructed a dataset called ROSE with retinal superficial and deep vascular annotations, respectively, to make up for the lack of datasets in this domain. The segmentation network OCTA-Net was also proposed to first generate an initial confidence map of blood vessels and then optimize the vessel details using a finetune module.

Among many deep learning-based medical image segmentation networks, the core ideology of U-Net [18] is the most prevalent, and our COSNet also takes advantage of it. The U-shaped structure facilitates the extraction of semantic information, while the skip connection contributes to restore positional information. In addition, the U-Net structure can be applied to all types of medical images virtually. In recent years, many variants based on the U-Net architecture have also emerged. Huang et al. [19] proposed a UNet3+ segmentation network based on full-scale jump connections and deep supervision. Full-scale skip connections combine low-level details with high-level semantics of feature maps at different scales, whereas deep supervision learns hierarchical representations from full-size aggregated feature maps. Isensee et al. [20] addressed the selection of pre-processing, architecture, training methods, and post-processing methods to propose a nn-UNet segmentation network that can automatically configure parameters, outperforming most methods on 23 international biomedical competition datasets. Tomar et al. [21] proposed the feedback attention network FANet for the network’s inability to efficiently learn information from different iteration batches, which combines feature maps from the current and previous batches together so that the masks of the previous batches constitute hard attention to learn the feature maps of different convolutional layers. Although the above methods achieve good results in image segmentation tasks, they focus only on intra- and inter-pixel semantic information of a single image, which prevents them from forming a structured feature space (large inter-class distance and small intra-class distance), thus limiting the segmentation performance.

Most current medical image segmentation methods rely entirely on accurate annotation, and medical image data are not as easily accessible as natural images. To address the data integration cost problem, researchers have introduced non-fully supervised methods to image segmentation tasks to reduce the reliance on annotation. In this paper, we focus on contrast learning, and the application of contrast learning to image segmentation is described below. Commonly used semantic segmentation methods for contrast learning are usually pre-trained on large unlabeled datasets first and then used for segmentation tasks, such as in the literature [22]. To solve the problem of training overfitting on a small amount of labeled data, Zhao et al. [23] proposed a new contrastive learning pre-training strategy, which is first pre-trained with label-based contrast loss and then fine-tuned with cross-entropy loss to improve intra-class compactness and inter-class separation. Zhong et al. [24] proposed a semi-supervised semantic segmentation method that both maintains the continuity of the prediction space for the input transform and also ensures the contrastive nature of the feature space. In contrast to the previous stepwise contrast learning segmentation, the contrast loss and semi-supervised segmentation training in this work are performed jointly. Alonso et al. [25] proposed a contrast learning module to perform pixel-level feature representation of similar samples from labeled and unlabeled data, and this work obtained good results in scenarios where the proportion of labeled data is small. Although all these methods noted above have achieved good results in the corresponding fields, they have not completely solved the problems of category imbalance and small target segmentation in medical image segmentation. COSNet in this paper also refers to similar loss function forms and utilizes the memory bank approach.

To address the problems of poor detail and class imbalance in small target segmentation tasks, we propose a global contrastive learning method for retinal OCTA image segmentation. Compared with previous methods, the method in this paper focuses on both single image information and global information (the whole dataset), which can achieve better results in a small sample medical dataset containing two kinds of annotations. The main contributions of this paper are as follows:A two-branch contrastive learning network for retinal OCTA image segmentation is proposed. The model is able to effectively extract features of vascular images by learning superficial and deep vascular annotations while avoiding the feature vanishing of deep vessels. A segmentation head and a projection head are added at the end of the decoder to obtain both segmentation mapping and pixel embedding;In this paper, a new pixel contrastive loss function is proposed. By saving same-class pixel queues and pixel embeddings in memory bank, features within a single image can be learned as well as same-class features in the whole dataset. The network model is guaranteed to learn more hard-to-segment samples, thus alleviating the class imbalance problem and improving the segmentation performance;A contrast-limited adaptive histogram equalization (CLANE) method with fixed area is used for retinal OCTA images to mitigate noise caused by imaging artifacts.

The subsequent sections of the text contain Methods and Theory, Experiments and Results, Discussion, and Conclusions.

## 2. Methods and Theory

We propose a fully-supervised contrastive learning model for retinal OCTA image segmentation called contrastive OCTA segmentation net (COSNet). This method is mainly divided into three parts: **feature extraction module**, **contrastive learning module,** and **fine-tune module**, as shown in Figure 1.

### 2.1. Feature Extraction Module

In the feature extraction module, a two-branch U-shaped network is adopted to handle data with two levels of annotation. This architecture has a partially shared “encoder + decoder” structure. The upper branch deals with superficial vessels (pixel-level label), and the lower branch deals with deep vessels (centerline-level), which can balance the importance of the two levels of information. Unlike previous segmentation networks, our method adds both a segmentation head and an MLP mapping head after the “encoder + decoder” structure. As shown in Figure 1, the ResNeSt module is used as the basic module for feature extraction, and the shallow vessel branch has five encoder layers and corresponding decoder layers, whereas the deep vessel branch has three encoder layers and corresponding decoder layers. Eventually, the segmentation mapping is obtained via a 1 × 1 convolutional layer, and the pixel embedding of the feature map is obtained by the MLP projection head behind the penultimate layer of the decoder.

As shown in Figure 2, we employ the ResNeSt [26] module as the basic unit, which serves the purpose of treating a series of representations as a combination of several feature groups and then using channel attention on these feature groups.

The variables *C*, *H*, and *W* represent channels, height, and width, respectively, of the feature map. The feature input is divided equally into *K* cardinal groups, which are then fed into *R* identical cardinal blocks. Thus, the initial input feature map is divided into *G* (*G* = *K* × *R*) groups along the channel dimension. Each branch contains the structure of “1 × 1 convolutional layer + BN + Relu” and “3 × 3 convolutional layer + BN + Relu”, resulting in a feature map with one-fourth of the original channels. In addition, each cardinal block is followed by a split attention to integrate feature map.

As shown in Figure 3, the feature map from two branches (*U*_1_ and *U*_2_) are fused, and then the channel-wise statistics *S* are generated via global pooling. It can be expressed as:(1)$$S_{c}^{k}=\frac{1}{H\times W}\sum_{i=1}^{H}\sum_{j=1}^{W}U_{c}^{k}\left(i,j\right).$$

The weighted fusion of the cardinal group representations is implemented using the channel dimension soft attention, where the feature maps of each channel are generated using a weighted combination. This process can be represented as:(2)$$V_{c}^{k}=\sum_{i=1}^{R}\alpha_{i}^{k}\left(c\right)U_{R(k-1))}.$$
where
(3)$$\alpha_{i}^{k}\left(c\right)=\left\{\begin{array}{cc} \frac{exp\left(G_{i}^{c}\left(s^{k}\right)\right)}{\sum_{j=0}^{R}exp\left(G_{i}^{c}\left(s^{k}\right)\right)} & R>1,\\ \frac{1}{1+exp\left(-G_{i}^{c}\left(s^{k}\right)\right)} & R=1. \end{array}\right.$$

Next, all the base arrays are concatenated to get the output, and by adding them up with shortcut τ(X), the total module output is *Y = V + τ(X)*.

### 2.2. Contrastive Learning Module

Currently, the loss functions used in most semantic segmentation methods focus only on local information in a single image and ignore the global context, i.e., same-class samples in the entire dataset. Examples include the cross-entropy loss function and the Dice loss function. In medical image segmentation networks, cross-entropy and Dice loss are usually used in a weighted manner to solve the problem of excessive oscillation of the loss function during training. However, the performance of such loss functions in segmentation becomes poor when the segmented target occupies a very small proportion of the full image. Considering the requirement of the loss function for detail difference sensitivity in the vessel segmentation task, we select the mean-square error loss (MSE) as part of the loss function. The mean-square loss function is defined as follows:(4)$$L_{MSE}=\frac{1}{n}\sum_{i=1}^{n}{\left(y_{pred}-y_{gt}\right)}^{2}.$$

To address the category imbalance problem, our approach borrows the idea of MoCo [27] to construct sample pairs, i.e., a memory bank. Unlike the MoCo method, the memory bank proposed in this paper stores a queue for each class. In each class, only a small number of pixels are randomly selected from the latest batch of images and are queued. At the same time, the same-class pixel embedding in each image is stored by mean-pooling. This approach achieves a better balance between learning speed and sample diversity than storing all pixel samples directly or updating only the last few batches. Specifically, suppose there are a total of *N* training images, *C* semantic classes, *D* is the dimension of the pixel embedding, and *T* is the length of the pixel queue, then the image pixel embedding size is *C × N × D*; the pixel queue size is *C × T × D*. The final size of the memory bank is *M = (N + T) × C × D*. The advantage of this approach is that more representative hard-to-segment samples could be stored, and memory consumption is lower.

The most vital step of contrast learning is how to reasonably select difficult samples. In OCTA images, difficult samples refer to those detailed parts of the deep vessels, where the width of the tiniest part is only one pixel. It can be further shown from Kalantidis et al. [28] that an increasing number of negative samples (background) over-simplifies the training, which leads to an ineffective penalty for contrastive loss. We use a hybrid strategy that includes hard sample mining and random sampling. The first few most difficult samples are sampled from the top of the memory bank using difficult sample mining, whereas the other half is sampled randomly. This hybrid sampling method allows pixel contrastive loss to focus more on the difficult pixels for segmentation.

The global pixel contrastive loss proposed in this paper is defined as follows:(5)$$L_{i}^{NCE}=\frac{1}{\left|P_{i}\right|}\sum_{i^{+}\in P_{i}}-log\frac{exp\left(i\cdot i^{+}/\tau \right)}{exp\left(i\cdot i^{+}/\tau \right)+\sum_{i^{-}\in N_{i}}exp\left(i\cdot i^{-}/\tau \right)}.$$
where $i^{+}$ and $i^{-}$ represent positive and negative samples, respectively; $P_{i}$ and $N_{i}$ represent the set of positive and negative pixel embeddings, respectively (positive and negative samples are from the entire dataset); “·” represents the inner product; τ represents a temperature hyperparameter. When pixel *i* belonging to class $\bar{C}$ is calculated, the pixel embedding of the same class $\bar{C}$ is considered as a positive sample and the other classes are considered as negative samples.

Therefore, combining the respective advantages of mean-square loss and global pixel contrastive loss, the objective function proposed is as follows:(6)$$L_{obj}=\sum_{i}\left(L_{MSE}+\lambda_{NCE}L_{i}^{NCE}\right).$$

Ultimately, both the superficial vascular complexes (SVC) branch and the deep vascular complexes (DVC) branch of COSNet utilize Equation (Equation 6) as the objective function, so the segmentation loss function equation of this module is defined as follows:(7)$$L_{total}=\lambda_{SVC}L_{SVC}+\lambda_{DVC}L_{DVC}.$$

### 2.3. Fine-Tune Module

To address the problem of detailed breakpoints in vessel segmentation, we directly follow the fine-tune module in OCTA-Net [17], as shown in Figure 4.

First, the SVC prediction map, DVC prediction map, and original image (all single channel) obtained from the previous module are channel-fused into a 3-channel input. The input passes through two and three convolutional layers (DVC and SVC), respectively, and then this module generates a normalized local propagation coefficient mapping of size m×m, for all positions of the output of the convolutional layers, denoted as Equation (Equation 8), where $h_{i}^{P}$ denotes the confidence value of neighbor *P* at position *i*, and m×m is the size of the propagating neighbor.
(8)$$\omega_{i}^{P}=\frac{exp\left(h_{i}^{P}\right)}{\sum_{t=1}^{m\times m}exp\left(h_{i}^{P}\right)},P\in \left[1,m\times m\right].$$

Local propagation coefficient $\omega_{i}^{P}$ of position *i* is multiplied by SVC and DVC confidence maps from the previous module, then aggregated to the centroid to yield fine-tuned result, denoted as Equation (Equation 9):(9)$$Pred_{i}=\sum_{P=1}^{m\times m}\omega_{i}^{P}\cdot f_{i}^{P}$$
where $Pred_{i}$ is the final prediction vector for position *i*, and $f_{i}^{P}$ is the confidence vector for neighbor *P* at position *i*. Finally, the fine-tuned SVC segmentation image and the DVC segmentation image are combined into a binary image.

In the fine-tuning module, we set the size of the aggregated neighbor coefficient *m* to 3 while using Dice loss as the fine-tune loss function. Dice loss function is denoted as Equation (Equation 10), where $P_{i}$, $G_{i}$ represent the prediction and ground truth, respectively. In addition, a small positive constant σ is used to avoid the numerical problem and accelerate convergence of training.
(10)$$L_{Dice}=1-\frac{2\sum_{i=1}^{N}P_{i}G_{i}+\sigma }{\sum_{i=1}^{N}P_{i}^{2}+\sum_{i=1}^{N}G_{i}^{2}+\sigma }.$$

## 3. Experiments and Results

In this section, we perform a comparison test and an ablation study in order to verify the effectiveness of COSNet on retinal vessel segmentation tasks (class-imbalanced and complex-structured vessel).

### 3.1. Experimental Configuration

#### 3.1.1. Dataset and Augmentation

The ROSE dataset is the first open-source OCTA dataset for segmentation, constructed by the Ningbo Institute of Materials affiliated with the Chinese Academy of Sciences, Southern University of Science and Technology, University of Liverpool, UK, IIAI, University of Southern California, and several research and clinical institutions in China and abroad. The ROSE dataset consists of two subsets named ROSE-1 and ROSE-2, which, in total, contain 229 OCTA images. The OCTA-500 dataset [29], built by Qiang Chen’s group from Nanjing University of Science and Technology, contains multimodal image data and multiple labels and different depth OCTA proiection. All datasets are divided into a training set and a validation set in the ratio of 7:3. The details of the datasets are shown in Table 1 and Table 2.

In addition to using common augmentation methods (e.g., random cropping, rotation, etc.), we employ a contrast-constrained adaptive histogram equalization (CLANE) method for preprocessing images in a fixed region. This method could enhance the detail features in DVC while reducing the noise generated by artifact problem. As can be seen in Figure 5, the DVC details in the green box are enhanced.

#### 3.1.2. Development Environment, Parameter Configuration, and Evaluation Metrics

The experimental hardware environment is an NVIDIA A40 graphics card, 80 GB running memory, and an AMD EPYC 7543 processor on AutoDl-GPU server; the software environment is the Ubantu18 system, Python 3.6.5, Pytorch 1.8.2 framework. Our COSNet method trains 200 epochs in both the segmentation and fine-tune stages, with the following settings: batch size of 16, temperature τ of 0.07, memory bank size of 400, Adam optimization with the initial learning rate of 0.0005, and weight decay of 0.0001. To speed up training, COSNet makes use of the ResNeSt50 pre-trained weight.

The following six metrics were tested: area under the curve (AUC), accuracy (ACC), G-mean, Kappa, Dice, and false discovery rate (FDR). They are defined in Equations (11)–(16), where *P* stands for positive and *N* for negative.
(11)$$\mathrm{AUC}=\frac{\sum {\left(p_{i},n_{j}\right)}_{p_{i}>n_{j}}}{P\times N}.$$
(12)$$\mathrm{ACC}=\frac{TP+TN}{TP+TN+FP+FN}.$$
(13)$$G\text{-}\mathrm{mean}=\sqrt{\frac{TP\times TN}{\left(TP+FN\right)\left(TN+FP\right)}}.$$
(14)$$\mathrm{Kappa}=\frac{ACC-p_{e}}{1-p_{e}},p_{e}=\frac{\left(TP+FN\right)\left(TP+FP\right)+\left(TN+FP\right)\left(TN+FN\right)}{{\left(TP+TN+FP+FN\right)}^{2}}.$$
(15)$$\mathrm{Dice}=\frac{2\times TP}{FP+FN+2\times TP}.$$
(16)$$\mathrm{FDR}=\frac{FP}{TP+FP}.$$

### 3.2. Comparison Test

To validate the superiority of COSNet on OCTA image segmentation tasks, COSNet is compared with existing methods on ROSE and OCTA-500. To ensure fairness, batch size is set to 16 and epoch of 400 (except COSFIRE), as listed in Table 3, Table 4, Table 5 and Table 6.

Except for FDR in Table 3, COSNet obtained the best performance. Although COSFIRE achieved the best score on the FDR metric, it was significantly lower than the deep learning method on other metrics. The filter-based method worked better on the simple structure of vessels. However, in the ROSE dataset, the superficial and deep microvascular structures are highly dense and intricate, which makes the segmentation effect of the traditional method poorer. In addition, the artifacts, low resolution, and low contrast of OCTA imaging also increase segmentation difficulty. In contrast, deep learning methods can extract higher-level distinguishing representations from local and global features, resulting in better performance on segmentation tasks. To visualize segmentation performance of different methods, we show the ROC curves for ROSE-1 and ROSE-2 in Figure 6. The comparison of segmentation results is also shown in Figure 7 and Figure 8. (We mainly focus on tasks containing two kinds of annotations, so only the results of ROSE-1 are shown).

Deep learning-based segmentation methods are superior to traditional methods such as COSFIRE for the OCTA image segmentation task. The basic structure of these segmentation networks is “encoder + decoder”, which is able to extract the deep semantic information. In addition, the use of various attention blocks can also help to improve the AUC score, such as the feedback attention mechanism of FANet. The COSNet proposed in this paper outperforms other approaches not only by using the ResNeSt block as the backbone of the “encoder + decoder” structure, but also by adopting a new pixel contrast loss function, which improves the information extraction ability of the network under the scenario of class imbalance. In addition, the fine-tune module can refine position information of DVC. From the green box in Figure 7 and Figure 8, it is easy to see that the previous methods have more breakpoints at deep vascular locations. In contrast, COSNet outperforms the other methods in the scenario of complex structure and class imbalance of microvessels.

### 3.3. Ablation Study

The ablation experiments in this subsection mainly explore the performance of different combinations of methods on ROSE-1. The role of skip connection has been verified many times in previous studies, so no separate experiments are done. Each set of experiments uses pre-training weight of ResNest50 and trains 400 epochs separately to record the highest score.

As shown in Table 7, we conducted ablation experiments on the COSNet architecture. Due to the form of the mean square loss, the difference between the predicted and true values are amplified, forming a strong supervision for the training, which is superior to the combination of cross entropy and Dice loss. In particular, our global contrastive loss function improves segmentation AUC by nearly 2 percentage points; other methods have less impact. This shows that our proposed combination of methods (Method 6) performs better on the ROSE dataset.

## 4. Discussion

Experimental results show that our COSNet is generally superior to existing methods. In the case of small segmentation targets and class imbalance, methods that employ only strongly supervised loss functions (e.g., U-Net) usually perform poorly. The reason is that the strongly supervised loss pays attention only to information in single images, but the classes in a single image are inherently in an unbalanced state, which leads to the model inadequately learning information of minority class (vessels).

We believe that the essence of image segmentation lies in constructing a structured feature space (smaller intra-class distance and larger inter-class distance). Therefore, in the network architecture, our method extracts both local information (single images) and global information (same-class samples), which effectively alleviates class imbalance. Due to the characteristics of contrast learning, categorization by cosine similarity, distinctive target (vessels) and background are easily segmented. The segmentation results could be utilized for clinical diagnosis by the ratio of superficial and deep vessels width (e.g., Alzheimer’s disease). The previous method was only designed for processing images containing superficial vascular annotations, so the clinical application is less extensive.

The experiments also find that using different sampling strategies has a great impact on the results, which also validates previous studies on contrast learning. The selection of sampling strategies needs to be further investigated with regard to the features of data. In the fine-tune stage, the breakpoint problem has not been completely solved, so there is still potential for improvement.

## 5. Conclusions

In summary, we propose a segmentation method based on global contrast learning for segmenting complex structured OCTA images, which includes a two-branch network architecture and a new pixel contrastive learning function. The core idea is to enable neural networks to learn similar features from a single image and the whole dataset simultaneously. In the scene of class imbalance, our method results in better segmentation accuracy. At the same time, COSNet is also transferable to other class-imbalanced classification or segmentation tasks. The segmentation results of retinal vascular plexus are available for the analysis and prediction of ocular related diseases, and some recent studies [30,31,32] have been published.

However, the resultant breakpoint of complex-structured vessel segmentation is still a problem that needs to be solved. We hope that more researchers will choose to open source their results in the future to promote the development of medical images.

## Figures and Tables

**Figure 1 sensors-22-09847-f001:**
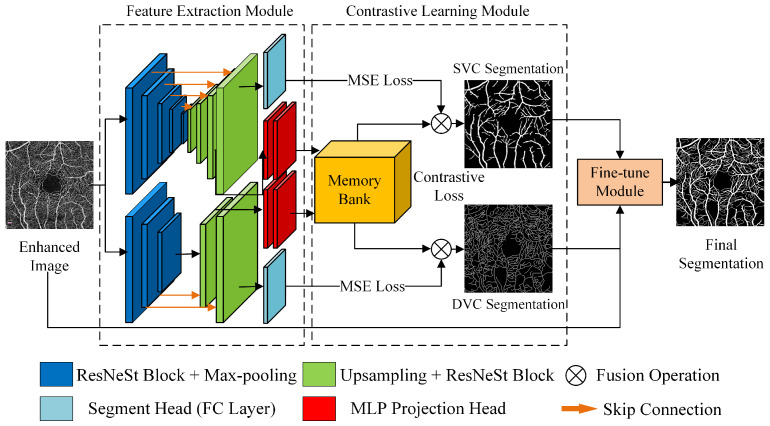
COSNet architecture.

**Figure 2 sensors-22-09847-f002:**
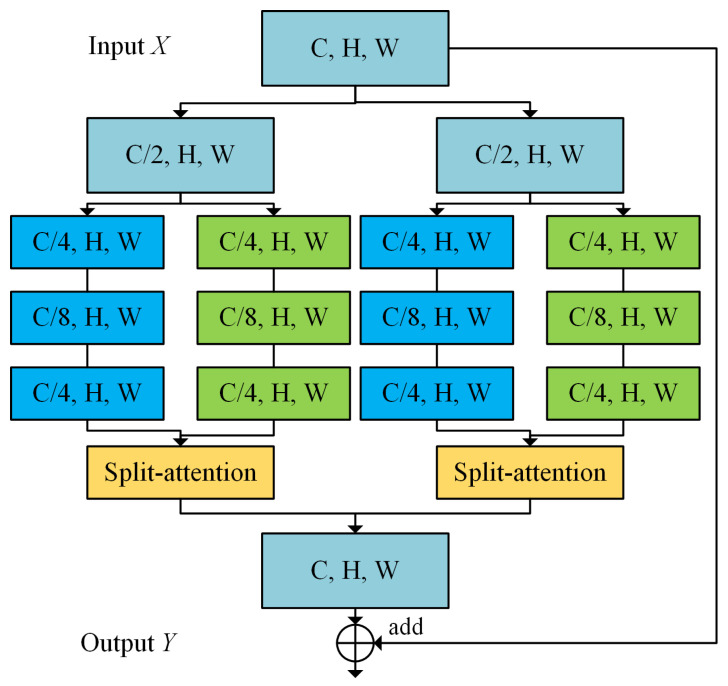
ResNeSt block (only two branches are shown for brevity).

**Figure 3 sensors-22-09847-f003:**
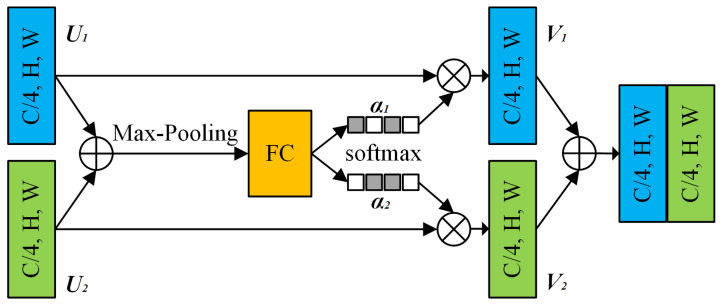
Split-attention block (only two branches are shown for brevity).

**Figure 4 sensors-22-09847-f004:**
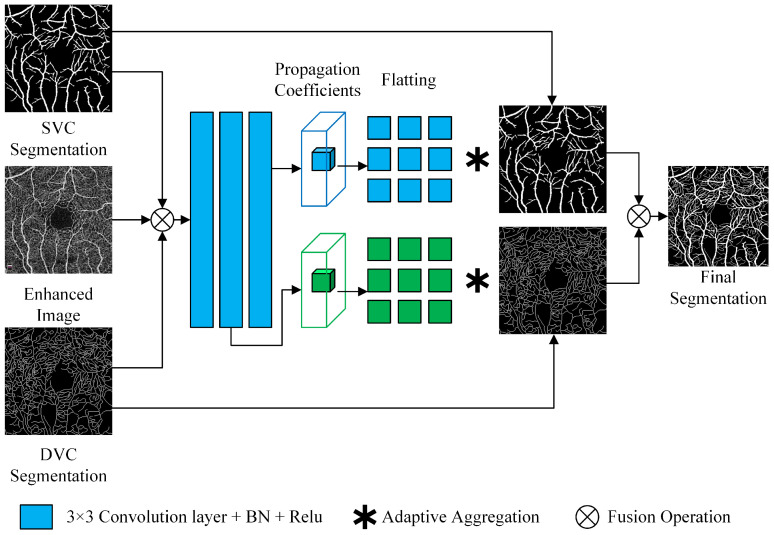
Fine-tune block.

**Figure 5 sensors-22-09847-f005:**
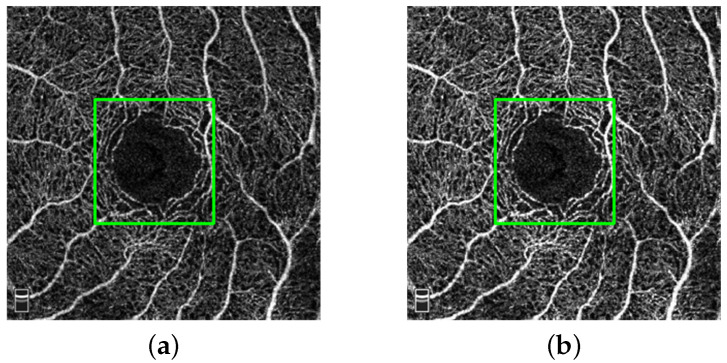
Original image (**a**) and enhanced image (**b**).

**Figure 6 sensors-22-09847-f006:**
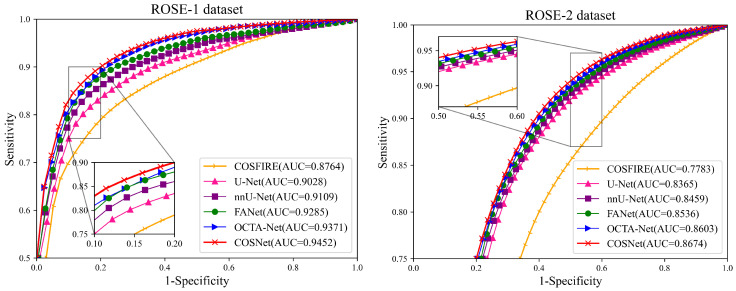
ROC curves for ROSE-1 (**left**) and ROSE-2 (**right**).

**Figure 7 sensors-22-09847-f007:**
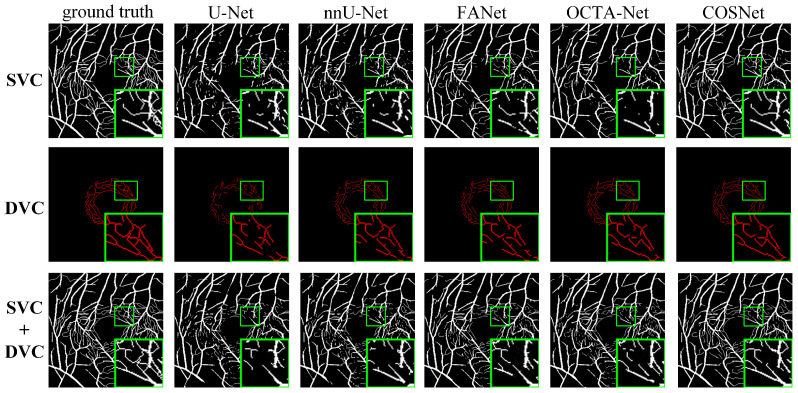
Segmentation results for ROSE-1.

**Figure 8 sensors-22-09847-f008:**
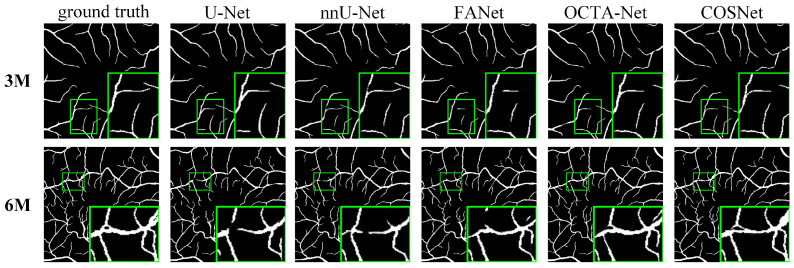
Segmentation results for OCTA-500 (ILM-OPL).

**Table 1 sensors-22-09847-t001:** Details of the ROSE dataset.

	ROSE-1	ROSE-2
Acquisition device	Optovue, USA	Heidelberg OCT2, Germany
Number	117	112
Resolution	304 × 304	512 × 512
Image type	SVC, DVC, SVC + DVC ^1^	SVC
Annotation type	pixel and centerline level	centerline level
Disease type	Alzheimer’s disease, macular degeneration, glaucoma, etc.	macular degeneration

^1^ SVC stands for superficial vascular complex angiography; DVC stands for deep vascular complex angiography.

**Table 2 sensors-22-09847-t002:** Details of the OCTA-500 * dataset.

	OCTA-3M	OCTA-6M
Number	200	300
Resolution	304 × 304	400 × 400
Image type	SVC	SVC
Annotation type	pixel level	pixel level

* OCTA-500 contains images of different depths (FULL, ILM-OPL, and OPL-BM); we use only the ILM-OPL data.

**Table 3 sensors-22-09847-t003:** Performance comparison on ROSE-1.

Method	AUC	ACC	G-Mean	Kappa	Dice	FDR
COSFIRE [6]	0.8764	0.8978	0.7253	0.6125	0.6673	**0.0985**
U-Net [18]	0.9028	0.8859	0.8038	0.6310	0.7015	0.2889
nnU-Net [20]	0.9109	0.8996	0.8185	0.6687	0.7311	0.2253
FANet [21]	0.9285	0.9057	0.8223	0.6815	0.7406	0.2230
OCTA-Net [17]	0.9371	0.9098	0.8335	0.7022	0.7570	0.2045
COSNet (Ours)	**0.9452**	**0.9133**	**0.8402**	**0.7097**	**0.7645**	0.2013

**Table 4 sensors-22-09847-t004:** Performance comparison on ROSE-2 *.

Method	AUC	ACC	G-Mean	Kappa	Dice	FDR
COSFIRE [6]	0.7783	0.9210	0.7745	0.5698	0.6143	0.3890
U-Net [18]	0.8365	0.9317	0.8000	0.6174	0.6559	0.3546
nnU-Net [20]	0.8459	0.9342	0.8077	0.6346	0.6696	0.3328
FANet [21]	0.8536	0.9373	0.8214	0.6578	0.6935	0.3236
OCTA-Net [17]	0.8603	0.9386	0.8313	0.6721	0.7078	0.3018
COSNet (Ours)	**0.8674**	**0.9398**	**0.8386**	**0.6738**	**0.7104**	**0.3002**

* Because ROSE-2 contains only centerline-level annotations, COSNet uses only the DVC branch.

**Table 5 sensors-22-09847-t005:** Performance comparison on OCTA-3M (ILM-OPL).

Method	AUC	ACC	G-Mean	Kappa	Dice	FDR
COSFIRE [6]	0.8542	0.8645	0.7544	0.6583	0.7402	0.2213
U-Net [18]	0.9156	0.9001	0.8021	0.6847	0.8054	0.2394
nnU-Net [20]	0.9358	0.9089	0.8087	0.7065	0.8365	0.2175
FANet [21]	0.9520	0.9275	0.8322	0.7344	0.8556	0.2160
OCTA-Net [17]	0.9524	0.9246	0.8457	0.7857	0.9085	0.2037
COSNet (Ours)	**0.9676**	**0.9345**	**0.8628**	**0.7992**	**0.9168**	**0.1857**

**Table 6 sensors-22-09847-t006:** Performance comparison on OCTA-6M (ILM-OPL).

Method	AUC	ACC	G-Mean	Kappa	Dice	FDR
COSFIRE [6]	0.8248	0.8392	0.7406	0.6084	0.7078	0.2859
U-Net [18]	0.8876	0.8802	0.8085	0.7158	0.8045	0.3189
nnU-Net [20]	0.8965	0.8854	0.8196	0.7295	0.8158	0.3064
FANet [21]	0.9057	0.8935	0.8172	0.7354	0.8365	0.2930
OCTA-Net [17]	0.9196	0.9107	0.8349	0.7536	0.8754	0.2778
COSNet (Ours)	**0.9388**	**0.9253**	**0.8457**	**0.7768**	**0.8869**	**0.2874**

**Table 7 sensors-22-09847-t007:** Ablation experiment on ROSE-1.

Method	AUC	ACC	G-Mean	Kappa	Dice	FDR
1 ^1^	0.8874	0.8645	0.7937	0.6673	0.7284	0.2524
2 ^2^	0.9199	0.9064	0.8285	0.6889	0.7463	0.2114
3 ^3^	0.9307	0.9092	0.8344	0.7002	0.7561	0.2135
4 ^4^	0.9321	0.9110	0.8375	0.7053	0.7604	0.2057
5 ^5^	0.9304	0.9002	0.8285	0.6974	0.7552	0.2103
6 ^6^	**0.9452**	**0.9133**	**0.8402**	**0.7097**	**0.7645**	**0.2013**

^1^ Cross-entropy loss + Dice loss. ^2^ Mse loss. ^3^ Mse loss + contrastive loss. ^4^ Mse loss + contrastive loss + image augmentation (CLANE). ^5^ Cross-entropy loss + Dice loss + contrastive loss + image augmentation (CLANE) + fine-tune. ^6^ Mse loss + contrastive loss + image augmentation (CLANE) + fine-tune.

## Data Availability

ROSE dataset link: https://imed.nimte.ac.cn/dataofrose.html; OCTA-500 dataset link: https://ieee-dataport.org/open-access/octa-500; Download project code at https://github.com/Pantherk1/COSNet.

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
