# Peer review of "Retinal OCTA Image Segmentation Based on Global Contrastive Learning"

_sensors, 2022, doi:10.3390/s22249847_

Round 1

Reviewer 1 Report

Please see attachment for comments.

Author Response

Dear Reviewer:

Thanks for your comments concerning our manuscript (ID: sensors-2057865). Those comments are valuable for revising and improving our paper with important guiding significance. We have made correction according to the comments and added the OCTA-500 dataset as a supplement to verify the effectiveness of COSNet. The responds to the reviewer's comments are as follows:

1.Please modify key words, about 3-5 words.

The number of keywords has been reduced to 5.

2.Please add a paragraph in introduction introducing the structure of this paper.

We have added an brief introduction to the structure of article at the end of section 1.

3.When the authors cite references, most of them are simply displayed in the paper. I suggest that the authors relate these references to the work of this paper, for example, how certain studies have influenced their work.

We have made the appropriate changes.

4.Please check the full text to ensure that every symbol is explained.

We have modified the image and also updated the abbreviations at the end of article.

5.Is the data set sufficient?

We added the OCTA-500 dataset as a supplement to verify the effectiveness of COSNet.

6.Please explain the principles and procedure of parameter setting.

After several experiments, we found that batchsize did not have much effect on the results, so we adjusted it to the upper limit to speed up the training;

For the weight decay and initial learning rate, we mainly refer to the settings of OCTA-Net, which are also more commonly used;

For the setting of temperature τ, we mainly referred to the following article and conducted experiments to select the optimal parameters.
https://doi.org/10.48550/arXiv.1911.05722

2004.11362.pdf (arxiv.org)

7.My suggestion is to add a paragraph before section 3.1 to introduce the main

content of section 3.

We have added a small paragraph.

8.Explain the evaluation metrics and justify why those evaluation metrics are used.

AUC is the area under the ROC curve, an evaluation metric that measures a binary classification and indicates the probability that predicted positive sample ranks ahead of a negative sample;

ACC is the percentage of the correct predictions;

G-mean is an integrated metric of positive accuracy and negative accuracy. Details can be found at (Kubat M, Holte R C, Matwin S. Machine Learning for the Detection of Oil Spills in Satellite Radar Images[J]. Machine Learning, 1998, 30(2-3):195-215.);

The Kappa coefficient is a metric used for consistency testing and can also be used to measure the effectiveness of classification. Consistency refers to whether the predicted results of the model are consistent with the actual classification results;

Dice coefficient is an ensemble similarity measure function, which is usually used to calculate the similarity of two samples and takes the value of [0,1];

False discovery rate (FDR) means "the percentage of all discoveries that have errors", which is important for measuring segmentation results.

9.My suggestion is to add data to illustrate the experimental results and discussion.

We added the OCTA-500 and conducted comparative tests.

10.Is this method useful in clinical practice?

I previously consulted with attending physician in Beijing Aier Intech Eye Hospital. She said the segmentation results of the vascular plexus are informative for the prevention and diagnosis of ocular related diseases.

11.What is the limitations of this method?

We believe that segmentation results are not good enough, and better post-segmentation processing methods are still needed to solve the breakpoint problem.

12.What are the further research topics and directions? What else can be improved?

Our further work is to investigate better segmentation methods and to predicting diseases based on the segmentation results.

Reviewer 2 Report

Please see the Attached review report.

Author Response

Dear Reviewer:

Thanks for your comments concerning our manuscript (ID: sensors-2057865). Those comments are valuable for revising and improving our paper with important guiding significance. We have made correction according to the comments and added the OCTA-500 dataset as a supplement to verify the effectiveness of COSNet. The responds to the reviewer's comments are as follows:

1.Structure of sentences need to be polished. Readiability is poor. Typographical mistakes are also there.

We have embellished the relevant statements, especially the introduction. Typographical errors have also been corrected.

2.Fusion module in Figure 1 and Figure 2 needs more clarity.

We have enlarged the image dpi and font size to make it clearer.

3.On page 4, "Thus the initial input feature map is divided into G = K × R..." This statement does not show the division mechanism. Instead it looks product group.

We have revised the relevant expressions to avoid ambiguity.

4.In Figure 7, OCTA-Net and COSNet show almost similar qualitative description. This needs to be improved for significant difference.

We added OCTA-500 as a complementary experiment, modified the image and enlarged the green box to show.

Reviewer 3 Report

Dear Authors,

I think topical issues will be really interesting, especially for professionals in this field.

My minor comments:

1. Formulas are parts of a sentence. If a sentence ends with a formula, a point is placed after it, not a comma.

2. There are many places in the text that need to be corrected, e.g. in lines 44, 95, 111 and elsewhere. Section 2 must begin with a capital letter.

3. The impression is that the authors lack attentiveness.

After making adjustments according to the above comments, in my opinion, the paper could be published as it is.

Author Response

Dear Reviewer:

Thanks for your comments concerning our manuscript (ID: sensors-2057865). Those comments are valuable for revising and improving our paper with important guiding significance. We have made correction according to the comments and added the OCTA-500 dataset as a supplement to verify the effectiveness of COSNet. The responds to the reviewer's comments are as follows:

1.Formulas are parts of a sentence. If a sentence ends with a formula, a point is placed after it, not a comma.

We have made changes to the formulas and tables.

2.There are many places in the text that need to be corrected, e.g. in lines 44, 95, 111 and elsewhere. Section 2 must begin with a capital letter.

We have corrected the relevant errors.

3.The impression is that the authors lack attentiveness.

Thank you for your comment, we will definitely pay more attention to it.

Round 2

Reviewer 1 Report

1. Please add the full name of each abbreviation when it first appears.
Such as "COSNet" (line 6), "MLP" (line 8), "ACC" and "AUC" (line 13).
Please check all parts of the manuscript.
2. Please add some keywords. There should be at least 7 keywords in the
manuscript.
3. It is better to add more references related to this topic in the
introduction, which are published in recent years.
Some suggestions are listed below:
https://doi.org/ 10.1002/ett.4080

https://doi.org/ 10.1007/s13735-021-00218-1

https://doi.org/ 10.7717/peerj-cs.613

https://doi.org/ 10.3390/electronics11132012 

https://doi.org/ 10.1007/s11042-021-10942-9

https://doi.org/ 10.1016/j.bspc.2021.103261
4. In line 95, "Our COSNet" should be "COSNet in this paper"
5. In Line 103, it should be "a two-branch contrastive learning network
is proposed..."
6. The "Conclusion" part can be rewritten. First, the work summary
should be described more specifically; Second, lines 316-317 should be
supported with some references. 

Author Response

Dear  Reviewer:

Thanks for your comments concerning our manuscript (ID: sensors-2057865). We have made the appropriate changes to address the comments you raised.

1.Please add the full name of each abbreviation when it first appears.
Such as "COSNet" (line 6), "MLP" (line 8), "ACC" and "AUC" (line 13).
Please check all parts of the manuscript.

We have made additions.

2. Please add some keywords. There should be at least 7 keywords in the
manuscript.

Keywords have been added to 7.

3. It is better to add more references related to this topic in the
introduction, which are published in recent years.

We have added recent references.

4. In line 95, "Our COSNet" should be "COSNet in this paper"

We have modified it.

5. In Line 103, it should be "a two-branch contrastive learning network
is proposed..."

We have modified it.

6. The "Conclusion" part can be rewritten. First, the work summary
should be described more specifically; Second, lines 316-317 should be
supported with some references. 

We have rewritten the conclusion.